# A framework for Multi-A(rmed)/B(andit) Testing with Online FDR Control

**Fanny Yang**
Dept. of EECS, U.C. Berkeley
fanny-yang@berkeley.edu

**Aaditya Ramdas**
Dept. of EECS and Statistics, U.C. Berkeley
ramdas@berkeley.edu

**Kevin Jamieson**
Allen School of CSE, U. of Washington
jamieson@cs.washington.edu

**Martin Wainwright**
Dept. of EECS and Statistics, U.C. Berkeley
wainwrig@berkeley.edu

## Abstract

We propose an alternative framework to existing setups for controlling false alarms when multiple A/B tests are run over time. This setup arises in many practical applications, e.g. when pharmaceutical companies test new treatment options against control pills for different diseases, or when internet companies test their default webpages versus various alternatives over time. Our framework proposes to replace a sequence of A/B tests by a sequence of best-arm MAB instances, which can be continuously monitored by the data scientist. When interleaving the MAB tests with an online false discovery rate (FDR) algorithm, we can obtain the best of both worlds: low sample complexity and any time online FDR control. Our main contributions are: (i) to propose reasonable definitions of a null hypothesis for MAB instances; (ii) to demonstrate how one can derive an always-valid sequential $p$-value that allows continuous monitoring of each MAB test; and (iii) to show that using rejection thresholds of online-FDR algorithms as the confidence levels for the MAB algorithms results in both sample-optimality, high power and low FDR at any point in time. We run extensive simulations to verify our claims, and also report results on real data collected from the New Yorker Cartoon Caption contest.

## 1 Introduction

Randomized trials are the default option to determine whether potential improvements of an alternative method (e.g. website design for a tech company, or medication in clinical trials for pharmaceutical companies) are significant compared to a well-established default. In the applied domain, this is often colloquially referred to as *A/B testing* or *A/B/n testing* for several alternatives. The standard practice is to divert a small amount of the traffic or patients to the alternative and control. If an alternative appears to be significantly better, it is implemented; otherwise, the default setting is maintained.

At first glance, this procedure seems intuitive and simple. However, in cases where the aim is to optimize over one particular metric, one can do better. In particular, this common tool suffers from several downsides. (1) First, one may wish to allocate more traffic to a better treatment if it is clearly better. Yet typical A/B/n testing frameworks split the traffic uniformly over alternatives. Adaptive techniques should help to detect better alternatives faster. (2) Second, companies often desire to continuously monitor an ongoing A/B test as they may adjust their termination criteria as time goes by and possibly stop earlier or later than originally intended. However, this practice may result in many more false alarms if not properly accounted for. This is one of the reasons for the lack of reproducibility of scientific results, an issue recently receiving increased attention from the public media. (3) Third, the lack of sufficient evidence or an insignificant improvement of the metric may make it undesirable from a practical or financial perspective to replace the default. Therefore, when a

company runs hundreds to thousands of A/B tests within a year, ideally the number of statistically insignificant changes that it made should be small relative to the total number of changes made. While controlling the false alarm rate of each individual test does *not* achieve this type of *false discovery rate* (FDR) control, there are known procedures in the multiple testing literature that are tailored to this problem.

In this paper, we provide a novel framework that addresses the above shortcomings of A/B or A/B/n testing. The first concern is tackled by employing recent advances in adaptive sampling like the pure-exploration multi-armed bandit (MAB) algorithm. For the second concern, we adopt the notion of any-time $p$-values for guilt-free continuous monitoring. Finally, we handle the third issue using recent results in online FDR control. Hence the combined framework can be described as doubly-sequential (sequences of MAB tests, each of which is itself sequential). Although each of those problems has been studied in hitherto disparate communities, how to leverage the best of all worlds, if at all possible, has remained an open problem. The main contributions of this paper are in successfully merging these ideas in a meta framework and presenting the conditions under which it can be shown to yield near-optimal sample complexity and FDR control.

The remainder of this paper is organized as follows. In Section 2, we lay out the conceptual challenges that we address in the paper, and describe a meta-algorithm that combines adaptive sampling strategies with FDR control procedures. Section 3 is devoted to the description of a concrete procedure, along with some theoretical guarantees on its properties. In Section 4, we discuss some results of our extensive experiments on both simulated and real-world data sets available to us.

## 2    Formal experimental setup and a meta-algorithm

In this section provide a high-level overview of our proposed combined framework aimed at addressing the shortcomings mentioned in the introduction. A specific instantiation of this meta-algorithm along with detailed theoretical guarantees are specified in Section 3.

For concreteness, we refer to the system designer, whether a tech company or a pharmaceutical company, as a (data) scientist. We assume that the scientist needs to possibly conduct an infinite number of experiments sequentially, indexed by $j$. Each experiment has one default setting, referred to as the *control*, and $K = K(j)$ alternative settings, called the *treatments* or *alternatives*. The scientist must return one of the $K + 1$ options that is the "best" according to some predefined metric, before the next experiment is started. Such a setup is a simple mathematical model both for clinical trials run by pharmaceutical labs, and A/B/n testing used at scale by tech companies.

One full experiment consists of a sequence of steps. In each step, the scientist assigns a new person to one of the $K + 1$ options and observes an outcome. In practice, the role of the scientist could be taken by an adaptive algorithm, which determines the assignment at time step $j$ by careful consideration of all previous outcomes. Borrowing terminology from the multi-armed bandit (MAB) literature, we refer to each of the $K + 1$ options as an *arm*, and each assignment to arm $i$ is termed "pulling arm $i$". For concreteness, we assign the index 0 to the control arm and note that it is known to the algorithm. Furthermore, we assume that the observable metric from each pull of arm $i = 0, 1, \ldots, K$ corresponds to an independent draw from an unknown probability distribution with expectation $\mu_i$. In the sequel we use $\mu_{i_\star} := \max_{i=1,\ldots,K} \mu_i$ to denote the mean of the best arm. We refer the reader to Table 1 in Appendix A for a glossary of the notation used throughout this paper.

### 2.1    Some desiderata and difficulties

Given the setup above, how can we mathematically describe the guarantees that the companies might desire from an improved multiple-A/B/n testing framework? For which parts can we leverage known results and what challenges remain?

For the purpose of addressing the first question, let us adopt terminology from the hypothesis testing literature and view each experiment as a test of a *null hypothesis*. Any claim that an alternative arm is the best is called a *discovery*, and when such a claim is erroneous, it is called a false discovery. When multiple hypotheses are to be tested, the scientist needs to define the quantity it wants to control. While we may desire that the probability of even a single false discovery is small, this is usually far too stringent for a large and unknown number of tests and results in low power. For this reason, [1] proposed that it may be more useful to control the expected ratio of false discoveries to the total number of discoveries (called the False Discovery Rate, or *FDR* for short) or the ratio of expected number of false discoveries to the expected number of total discoveries (called the modified FDR

or *mFDR* for short). Over the past decades, the FDR and its variants like the mFDR have become standard quantities for multiple testing applications. In the following, if not otherwise specified, we use the term FDR to denote both measures in order to simplify the presentation. In Section 3, we show that both mFDR and FDR can be controlled for different choices of procedures.

### 2.1.1 Challenges in viewing an MAB instance as a hypothesis test

In our setup, we want to be able to control the FDR at any time in an online manner. Online FDR procedures were first introduced by Foster and Stine [2], and have since been studied by other authors (e.g., [3, 4]). They are based on comparing a valid $p$-value $P^j$ with carefully-chosen levels $\alpha_j$ for each hypothesis test[1]. We reject the null hypothesis, represented as $R_j = 1$, when $P^j \leq \alpha_j$ and we set $R_j = 0$ otherwise.

As mentioned, we want to use adaptive MAB algorithms to test each hypothesis, since they can find a best arm among $K + 1$ with near-optimal sample complexity. However the traditional MAB setup does not account for the asymmetry between the arms as is the case in a testing setup, with one being the default (control) and others being alternatives (treatments). This is the standard scenario in A/B/n testing applications, as e.g. a company might prefer wrong claims that the control is the best (false negative), rather than wrong claims that an alternative is the best (false positive), simply because new system-wide adoption of selected alternatives might involve high costs. What would be a suitable null hypothesis in this hybrid setting? For the sake of continuous monitoring, is it possible to define and compute always-valid $p$-values that are super-uniformly distributed under the null hypothesis when computed at any time $t$?

In addition to asymmetry, the practical scientist might have a different incentive than the ideal outcome for MAB algorithms as he/she might not want to find the best alternative if it is not *substantially* better than the control. Indeed, if the net gain is small, it might be offset by the cost of implementing the change from the existing default choice. By similar reasoning, we may not require identifying the single best arm if there is a *set* of arms with similar means all larger than the rest. We propose a sensible null-hypothesis for each experiment which incorporates the approximation and minimum improvement requirement as described above, and provide an always valid $p$-value which can be easily calculated at each time step in the experiment. We show that a slight modification of the usual LUCB algorithm caters to this specific null-hypothesis while still maintaining near-optimal sample complexity.

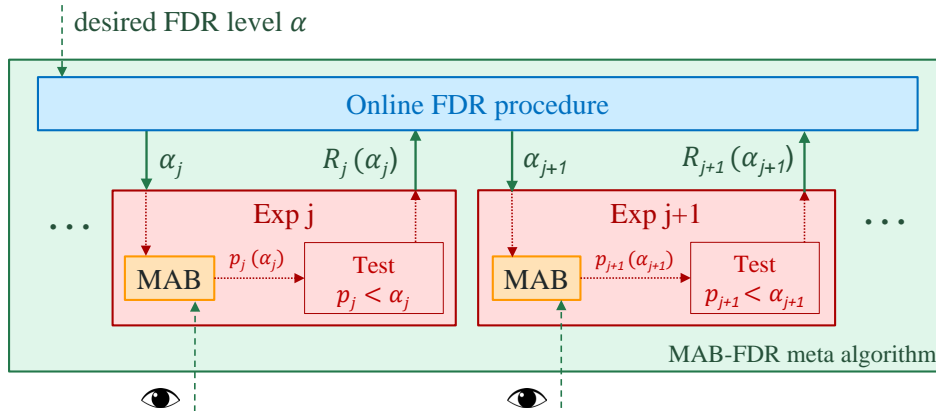

Figure 1: Diagram of our MAB-FDR meta algorithm. The green solid arrows symbolize interaction between the MAB and FDR procedures via the FDR test levels $\alpha_j$ and rejection indicator variables $R_j$. Notice that the $P^j$-values are now dependent as each $\alpha_j$ depends on the past rejections $R_1, \ldots, R_{j-1}$. The eyes represent possible continuous monitoring by the scientist.

### 2.1.2 Interaction between MAB and FDR

In order to take advantage of the sample efficiency of best-arm bandit algorithms, it is crucial to set the confidence levels close to what is needed. Given a user-defined level $\alpha$, at each hypothesis $j$, online

FDR procedures automatically output the significance level $\alpha_j$ which are sufficient to guarantee FDR control, based on past decisions. Can we directly set the MAB confidence levels to these output levels $\alpha_j$? If we do, our $p$-values are not independent across different hypotheses anymore: $P^j$ directly depends on the FDR levels $\alpha_j$ and each $\alpha_j$ in turn depends on past MAB rejections, thus on past MAB $p$-values (see Figure 1). Does the new interaction compromise FDR guarantees?

Although known procedures as in [2, 4] guarantee FDR control for independent $p$-values, this does not hold for dependent $p$-values in general. Hence FDR control guarantees cannot simply be obtained out of the box. A key insight that emerges from our analysis is that an appropriate bandit algorithm actually shapes the $p$-value distribution under the null in a "good" way that allows us to control FDR.

### 2.2 A meta-algorithm

Procedure 1 summarizes our doubly-sequential procedure, with a corresponding flowchart in Figure 1. We will prove theoretical guarantees after instantiating the separate modules. Note that our framework allows the scientist to plug in their favorite best-arm MAB algorithm or online FDR procedure. The choice for each of them determines which guarantees can be proven for the entire setup. Any independent improvement in either of the two parts would immediately lead to an overall performance boost of the overall framework.

---

**Procedure 1** MAB-FDR Meta algorithm skeleton

---

1. The scientist sets a desired FDR control rate $\alpha$.

2. For each $j = 1, 2, \ldots$:
   - Experiment $j$ receives a designated control arm and some number of alternative arms.
   - An *online-FDR procedure* returns an $\alpha_j$ that is some function of the past values $\{P^\ell\}_{\ell=1}^{j-1}$.
   - An *MAB procedure* is executed with inputs (a) the control arm and $K(j)$ alternative arms, (b) confidence level $\alpha_j$, maintains an always valid $p$-value for each $t$ and if the procedure self-terminates, returns a recommended arm.
   - When the MAB procedure is terminated at time $t$ by itself or the user, if the arm with the highest empirical mean is *not* the control arm and $P_t^j \leq \alpha_j$, then we return $P^j := P_t^j$, and the control arm is rejected in favor of this empirically best arm.

---

## 3 A concrete procedure with guarantees

We now take the high-level road map given in Procedure 1, and show that we can obtain a concrete, practically implementable framework with FDR control and power guarantees. We first discuss the key modeling decisions we have to make in order to seamlessly embed MAB algorithms into an online FDR framework. We then outline a modified version of a commonly used best-arm algorithm, before we finally prove FDR and power guarantees for the concrete combined procedure.

### 3.1 Defining null hypotheses and constructing $p$-values

Our first task is to define a null hypothesis for each experiment. As mentioned before, the choice of the null is not immediately obvious, since we sample from *multiple* distributions *adaptively* instead of independently. In particular, we will generally not have the same number of samples for all arms. Given a default mean $\mu_0$ and alternatives means $\{\mu_i\}_{i=1}^K$, we propose that the null hypothesis for the $j$-th experiment should be defined as

$$H_0^j : \mu_0 \geq \mu_i - \epsilon \quad \text{for all } i = 1, \ldots, K, \tag{1}$$

where we usually omit the index $j$ for simplicity. It remains to define an *always valid p-value* (previously defined by Johari et al. [5]) for each experiment for the purpose of continuous monitoring. It is defined as a stochastic process $\{P_t\}_{t=1}^\infty$ such that for all fixed and random stopping times $T$, under any distribution $\mathbb{P}_0$ over the arm rewards such that the null hypothesis is true, we have

$$\mathbb{P}_0(P_T \leq \alpha) \leq \alpha. \tag{2}$$

When all arms are drawn independently an equal number of times, by linearity of expectation one can regard the distance of each pair of samples as a random variable drawn i.i.d. from a distribution with mean $\tilde{\mu} := \mu_0 - \mu_i$. We can then view the problem as testing the standard hypothesis $H_0^j : \tilde{\mu} > -\epsilon$. However, when the arms are pulled adaptively, a different solution needs to be found—indeed, in this

case, the sample means are *not unbiased estimators* of the true means, since the number of times an arm was pulled now depends on the empirical means of all the arms.

Our strategy is to construct always valid $p$-values by using the fact that p-values can be obtained by inverting confidence intervals. To construct always-valid confidence bounds, we resort to the fundamental concept of the law of the iterated logarithm (LIL), for which non-asymptotic versions have been recently derived and used for both bandits and testing problems (see [6], [7]).

To elaborate, define the function

$$\varphi_n(\delta) = \sqrt{\frac{\log(\frac{1}{\delta}) + 3\log(\log(\frac{1}{\delta})) + \frac{3}{2}\log(\log(en))}{n}}. \tag{3}$$

If $\widehat{\mu}_{i,n}$ is the empirical average of independent samples from a sub-Gaussian distribution, then it is known (see, for instance, [8, Theorem 8]) that for all $\delta \in (0,1)$, we have

$$\max\left\{\mathbb{P}\Big(\bigcup_{n=1}^{\infty}\{\widehat{\mu}_{i,n} - \mu_i > \varphi_n(\delta \wedge 0.1)\}\Big),\quad \mathbb{P}\Big(\bigcup_{n=1}^{\infty}\{\widehat{\mu}_{i,n} - \mu_i < -\varphi_n(\delta \wedge 0.1)\}\Big)\right\} \leq \delta, \tag{4}$$

where $\delta \wedge 0.1 := \min\{\delta, 0.1\}$.

We are now ready to propose single arm $p$-values of the form

$$P_{i,t} := \sup\left\{\gamma \in [0,1] \mid \widehat{\mu}_{i,n_i(t)} - \varphi_{n_i(t)}(\tfrac{\gamma}{2K}) \leq \widehat{\mu}_{0,n_0(t)} + \varphi_{n_0(t)}(\tfrac{\gamma}{2}) + \epsilon\right\} \tag{5}$$

$$= \sup\left\{\gamma \in [0,1] \mid \mathrm{LCB}_i(t) \leq \mathrm{UCB}_0(t) + \epsilon\right\}$$

Here we set $P_{i,t} = 1$ if the supremum is taken over an empty set. Given these single arm $p$-values, the always-valid $p$-value for the experiment is defined as

$$P_t := \min_{s \leq t}\ \min_{i=1,\ldots,K} P_{i,s}. \tag{6}$$

We claim that this procedure leads to an always valid $p$-value (with proof in Appendix C).

**Proposition 1.** *The sequence $\{P_t\}_{t=1}^{\infty}$ defined via equation* (6) *is an always valid $p$-value.*

## 3.2 Adaptive sampling for best-arm identification

In the traditional A/B testing setting described in the introduction, samples are allocated uniformly to the different alternatives. But by allowing adaptivity, decisions can be made with the same statistical significance using far fewer samples. Suppose moreover that there is a unique maximizer $i_\star := \arg\max_{i=0,1,\ldots,K} \mu_i$, so that $\Delta_i := \mu_{i_\star} - \mu_i > 0$ for all $i \neq i_\star$. Then for any $\delta \in (0,1)$, best-arm MAB algorithms can identify $i_\star$ with probability at least $1-\delta$ based on at most[2] $\sum_{i \neq i_\star} \Delta_i^{-2}\log(1/\delta)$ total samples (see the paper [9] for a brief survey and [10] for an application to clinical trials). In contrast, if samples are allocated *uniformly* to the alternatives under the same conditions, then the most natural procedures require $K \max_{i \neq i_\star} \Delta_i^{-2}\log(K/\delta)$ samples before returning $i_\star$ with probability at least $1 - \delta$.

However, standard best-arm bandit algorithms do not incorporate asymmetry as induced by null-hypotheses as in definition (1) by default. Furthermore, recall that a practical scientist might desire the ability to incorporate approximation and a minimum improvement requirement. More precisely, it is natural to consider the requirement that the returned arm $i_b$ satisfies the bounds $\mu_{i_b} \geq \mu_0 + \epsilon$ and $\mu_{i_b} \geq \mu_{i_\star} - \epsilon$ for some $\epsilon > 0$. In Algorithm 1 we present a modified MAB algorithm based on the common LUCB algorithm (see [11, 12]) which incorporates the above desiderata. We provide a visualization of how $\epsilon$ affects the usual stopping condition in Figure 4 in Appendix A.1.

The following proposition applies to Algorithm 1 run with a control arm indexed by $i = 0$ with mean $\mu_0$ and alternative arms indexed by $i = 1, \ldots, K$ with means $\mu_i$, respectively. Let $i_b$ denote the random arm returned by the algorithm assuming that it exits, and define the set

$$\mathcal{S}^\star := \{i_\star \neq 0 \mid \mu_{i_\star} \geq \max_{i=1,\ldots,K} \mu_i - \epsilon \quad \text{and} \quad \mu_{i_\star} > \mu_0 + \epsilon\}. \tag{7}$$

**Algorithm 1** Best-arm identification with a control arm for confidence $\delta$ and precision $\epsilon \geq 0$
---
For all $t$ let $n_i(t)$ be the number of times arm $i$ has been pulled up to time $t$. In addition, for each arm $i$ let $\widehat{\mu}_i(t) = \frac{1}{n_i(t)} \sum_{\tau=1}^{n_i(t)} r_i(\tau)$, define

$$\text{LCB}_i(t) := \widehat{\mu}_{i,n_i(t)} - \varphi_{n_i(t)}(\tfrac{\delta}{2K}) \qquad \text{and} \qquad \text{UCB}_i(t) := \widehat{\mu}_{i,n_i(t)} + \varphi_{n_i(t)}(\tfrac{\delta}{2}).$$

1. Set $t = 1$ and sample every arm once.
2. Repeat: Compute $h_t = \arg\max_{i=0,1,\ldots,K} \widehat{\mu}_i(t)$, and $\ell_t = \arg\max_{i=0,1,\ldots,K, i\neq h_t} \text{UCB}_i(t)$

   (a) If $\text{LCB}_0(t) > \text{UCB}_i(t) - \epsilon$, for all $i \neq 0$, then output 0 and terminate.
   Else if $\text{LCB}_{h_t}(t) > \text{UCB}_{\ell_t}(t) - \epsilon$ and $\text{LCB}_{h_t}(t) > \text{UCB}_0(t) + \epsilon$, then output $h_t$ and terminate.

   (b) If $\epsilon > 0$, let $u_t = \arg\max_{i\neq 0} \text{UCB}_i(t)$ and pull all distinct arms in $\{0, u_t, h_t, \ell_t\}$ once. If $\epsilon = 0$, pull arms $h_t$ and $\ell_t$ and set $t = t + 1$.
---

Note that the mean associated with any index $i_\star \in \mathcal{S}^\star$, assuming that the set is non-empty, is guaranteed to be $\epsilon$-superior to the control mean, and at most $\epsilon$-inferior to the maximum mean over all arms.

**Proposition 2.** *The algorithm 1 terminates in finite time with probability one. Furthermore, suppose that the samples from each arm are independent and sub-Gaussian with scale* 1. *Then for any $\delta \in (0, 1)$ and $\epsilon \geq 0$, Algorithm 1 has the following guarantees:*

*(a) Suppose that $\mu_0 > \max_{i=1,\ldots,K} \mu_i - \epsilon$. Then with probability at least $1 - \delta$, the algorithm exits with*
$i_b = 0$ *after taking at most $O\left(\sum_{i=0}^{K} \widetilde{\Delta}_i^{-2} \log(K \log(\widetilde{\Delta}_i^{-2})/\delta)\right)$ time steps with effective gaps*

$$\widetilde{\Delta}_0 = (\mu_0 + \epsilon) - \max_{j=1,\ldots,K} \mu_j \quad and$$
$$\widetilde{\Delta}_i = (\mu_0 + \epsilon) - \mu_i.$$

*(b) Otherwise, suppose that the set $\mathcal{S}^\star$ as defined in equation* (7) *is non-empty. Then with probability at least $1 - \delta$, the algorithm exits with $i_b \in \mathcal{S}^\star$ after taking at most*
$O\left(\sum_{i=0}^{K} \widetilde{\Delta}_i^{-2} \log(K \log(\widetilde{\Delta}_i^{-2})/\delta)\right)$ *time steps with effective gaps*

$$\widetilde{\Delta}_0 = \min\left\{ \max_{j=1,\ldots,K} \mu_j - (\mu_0 + \epsilon), \max\{\Delta_0, \epsilon\} \right\} \quad and$$
$$\widetilde{\Delta}_i = \max\left\{ \Delta_i, \min\left\{ \max_{j=1,\ldots,K} \mu_j - (\mu_0 + \epsilon), \epsilon \right\} \right\}.$$

See Appendix D for the proof of this claim. Part (a) of Proposition 2 guarantees that when no alternative arm is $\epsilon$-superior to the control arm (i.e. under the null hypothesis), the algorithm stops and returns the control arm with probability at least $1 - \delta$. Part (b) guarantees that if there is in fact at least one alternative that is $\epsilon$-superior to the control arm (i.e. under the alternative), then the algorithm will find at least one of them that is at most $\epsilon$-inferior to the best of all possible arms.

As our algorithm is a slight modification of the LUCB algorithm, the results of [11, 12] provide insight into the number of samples taken before the algorithm terminates. Indeed, when $\epsilon = 0$ and $i_\star = \arg\max_{i=0,1,\ldots,K} \mu_i$ is a unique maximizer, the nearly optimal sample complexity result of [12] implies that the algorithm terminates under settings (a) and (b) after at most $\max_{j\neq i_\star} \Delta_j^{-2} \log(K \log(\Delta_j^{-2})/\delta) + \sum_{i\neq i_\star} \Delta_i^{-2} \log(\log(\Delta_i^{-2})/\delta)$ samples are taken (ignoring constants), where $\Delta_i = \mu_{i_\star} - \mu_i$.

In our development to follow, we now bring back the index for experiment $j$, in particular using $P^j$ to denote the quantity $P_T^j$ at any stopping time $T$. Here the stopping time can either be defined by the scientist, or in an algorithmic manner.

### 3.3 Best-arm MAB interacting with online FDR

After having established null hypotheses and $p$-values in the context of best-arm MAB algorithms, we are now ready to embed them into an online FDR procedure. In the following, we consider $p$-values for the $j$-th experiment $P^j := P^j_{T_j}$ which is just the $p$-value as defined in equation (6) at the stopping time $T_j$, which depends on $\alpha_j$.

We denote the set of true null and false null hypotheses up to experiment $J$ as $\mathcal{H}_0(J)$ and $\mathcal{H}_1(J)$ respectively, where we drop the argument whenever it's clear from the context. The variable $R_j = \mathbb{1}_{P^j \leq \alpha_j}$ indicates whether a the null hypothesis of experiment $j$ has been rejected, where $R_j = 1$ denotes a claimed discovery that an alternative was better than the control. The false discovery rate (FDR) and modified FDR *up to experiment $J$* are then defined as

$$\text{FDR}(J) := \mathbb{E}\frac{\sum_{j \in \mathcal{H}_0} R_j}{\sum_{i=1}^J R_i \vee 1} \qquad \text{and} \qquad \text{mFDR}(J) := \frac{\mathbb{E}\sum_{j \in \mathcal{H}_0} R_j}{\mathbb{E}\sum_{i=1}^J R_i + 1}. \qquad (8)$$

Here the expectations are taken with respect to distributions of the arm pulls and the respective sampling algorithm. In general, it is not true that control of one quantity implies control of the other. Nevertheless, in the long run (when the law of large numbers is a good approximation), one does not expect a major difference between the two quantities in practice.

The set of true nulls $\mathcal{H}_0$ thus includes all experiments where $H_0^j$ is true, and the FDR and mFDR are well-defined for any number of experiments $J$, since we often desire to control $\text{FDR}(J)$ or $\text{mFDR}(J)$ for all $J \in \mathbb{N}$. In order to measure power, we define the $\epsilon$-*best-arm discovery rate* as

$$\epsilon\text{BDR}(J) := \frac{\mathbb{E}\sum_{j \in \mathcal{H}_1} R_j \mathbb{1}_{\mu_{i_b} \geq \mu_{i_\star} - \epsilon} \mathbb{1}_{\mu_{i_b} \geq \mu_0 + \epsilon}}{|\mathcal{H}_1(J)|} \qquad (9)$$

We provide a concrete procedure 2 for our doubly sequential framework, where we use a particular online FDR algorithm due to Javanmard and Montanari [4] known as LORD; the reader should note that other online FDR procedure could be used to obtain essentially the same set of guarantees. Given a desired level $\alpha$, the LORD procedure starts off with an initial "$\alpha$-wealth" of $W(0) < \alpha$. Based on a inifinite sequence $\{\gamma_i\}_{i=1}^\infty$ that sums to one, and the time of the most recent discovery $\tau_j$, it uses up a fraction $\gamma_{j-\tau_j}$ of the remaining $\alpha$-wealth to test. Whenever there is a rejection, we increase the $\alpha$-wealth by $\alpha - W(0)$. A feasible choice for a stopping time in practice is $T_j := \min\{T(\alpha_j), T_S\}$, where $T_S$ is a maximal number of samples the scientist wants to pull and $T(\alpha_j)$ is the stopping time of the best-arm MAB algorithm run at confidence $\alpha_j$.

---

**Procedure 2** MAB-LORD: best-arm identification with online FDR control

1. Initialize $W(0) < \alpha$, set $\tau_0 = 0$, and choose a sequence $\{\gamma_i\}$ s.t. $\sum_{i=1}^\infty \gamma_i = 1$
2. At each step $j$, compute $\alpha_j = \gamma_{j-\tau_j} W(\tau_j)$ and
   $W(j+1) = W(j) - \alpha_j + R_j(\alpha - W(0))$
3. Output $\alpha_j$ and run Algorithm 1 using $\alpha_j$-confidence and stop at a stopping time $T_j$.
4. Algorithm 1 returns $P^j$ and we reject the null hypothesis if $P^j \leq \alpha_j$.
5. Set $R_j = \mathbb{1}_{P^j \leq \alpha_j}, \tau_j = \tau_{j-1} \vee jR_j$, update $j = j + 1$ and go back to step 2.

---

The following theorem provides guarantees on mFDR and power for the MAB-LORD procedure.

**Theorem 1** (Online mFDR control for MAB-LORD)**.**

*(a) Procedure 2 achieves mFDR control at level $\alpha$ for stopping times $T_j = \min\{T(\alpha_j), T_S\}$.*

*(b) Furthermore, if we set $T_S = \infty$, Procedure 2 satisfies*

$$\epsilon BDR(J) \geq \frac{\sum_{j=1}^J \mathbb{1}_{j \in \mathcal{H}_1}(1 - \alpha_j)}{|\mathcal{H}_1(J)|}. \qquad (10)$$

See Appendix E for the proof of this claim. Note that by the arguments in the proof of Theorem 1, mFDR control itself is actually guaranteed for any generalized $\alpha$-investing procedure [3] combined with any best-arm MAB algorithm. In fact we could use any adaptive stopping time $T_j$ which depend on the history only via the rejections $R_1, \ldots, R_{j-1}$. Furthermore, using a modified LORD proposed

by Javanmard and Montanari [13], we can also guarantee FDR control– a result we moved to the Appendix F due to space constraints. It is noteworthy that small values of $\alpha$ do not only guarantee smaller FDR error but also higher BDR. However, there is no free lunch — a smaller $\alpha$ implies a smaller $\alpha_j$ at each experiment, resulting in a larger required number of pulls for the the best-arm MAB algorithm.

# 4 Experimental results

In the following, we briefly describe some results of our experiments[3] on both simulated and real-world data sets, which illustrate that, apart from FDR control, MAB-FDR (used interchangeably with MAB-LORD here) is highly advantageous in terms of sample complexity and power compared to a straightforward embedding of A/B testing in online FDR procedures. Unless otherwise noted, we set $\epsilon = 0$ in all of our simulations to focus on the main ideas and keep the discussion concise.

**Competing procedures** There are two natural frameworks to compare against MAB-FDR. The first, called AB-FDR or AB-LORD, swaps the MAB part for an A/B (i.e. A/B/n) test (uniformly sampling all alternatives until termination). The second comparator exchanges the online FDR control for independent testing at $\alpha$ for all hypotheses – we call this MAB-IND. Formally, AB-FDR swaps step 3 in Procedure 2 with "*Output $\alpha_j$ and uniformly sample each arm until stopping time $T_j$.*" while MAB-IND swaps step 4 in Procedure 2 with "*The algorithm returns $P^j$ and we reject the null hypothesis if $P^j \leq \alpha$.*". In order to compare the performances of these procedures, we ran three sets of simulations using Procedure 2 with $\epsilon = 0$ and $\gamma_j = 0.07 \frac{\log(j \vee 2)}{j e^{\sqrt{\log j}}}$ as in [4].

Our experiments are run on artificial data with Gaussian/Bernoulli draws and real-world Bernoulli draws from the New Yorker Cartoon Caption Contest. Recall that the sample complexity of the best-arm MAB algorithm is determined by the gaps $\Delta_j = \mu_{i_\star} - \mu_j$. One of the main relevant differences to consider between an experiment of artificial or real-world nature is thus the distribution of the means $\mu_i$ for $i = 1, \ldots, K$. The artificial data simulations are run with a fixed gap $\Delta := \Delta_2$ while the means of the other arms are set uniformly in $[0, \mu_{i_\star} - \Delta]$. For our real-world simulations, we use empirical means computed from the cartoon caption contest (see details in Appendix B.1.1). In addition, the contests actually follow a natural chronological order, which makes this dataset highly relevant to our purposes. In all simulations, 60% of all the hypotheses are true nulls, and their indices are chosen uniformly. Due to space constraints, the experimental results for artificial and real-world Bernoulli draws are deferred to Appendix B.

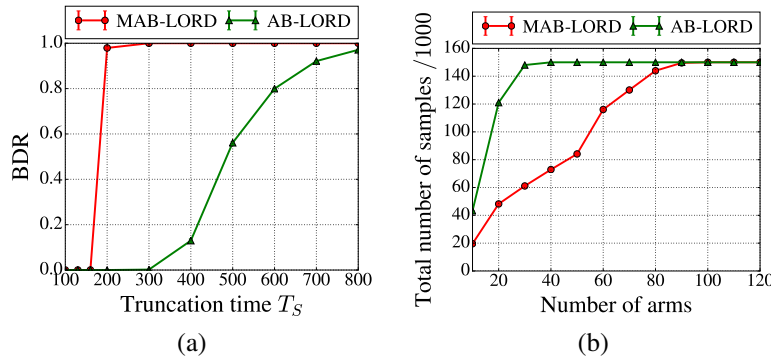

Figure 2: (a) Power vs. truncation time $T_S$ (per hypothesis) for 50 arms and (b) Sample complexity vs. # arms for truncation time $T_S = 300$ for Gaussian draws with fixed $\mu_{i_\star} = 8$, $\Delta = 3$ over 500 hypotheses with 200 non-nulls, averaged over 100 runs and $\alpha = 0.1$.

**Power and sample complexity** In this section we include figures on artificial Gaussian trials which confirm that the total number of necessary pulls to determine significance is much smaller for MAB-FDR than for AB-FDR. In Fig. 2 (a) we fix the number of arms and plot the $\epsilon$BDR with $\epsilon = 0$ (BDR for short) for both procedures over different choices of truncation times $T_S$. Low BDR indicates that the algorithm often reaches truncation time before it could stop. For Fig. 2 (b) we fix $T_S$ and show how the sample complexity varies with the number of arms.

Observe in Fig. 2 (a) that the power at any given truncation time is much higher for MAB-FDR than AB-FDR. This is because the best-arm MAB is more likely to satisfy the stopping criterion before any given truncation time than the uniform sampling algorithm. Fig. 2(b) qualitatively shows how the total number of necessary arm pulls for AB-FDR increases much faster with the number of arms than for MAB-FDR before it plateaus due to the truncation. Recall that whenever the best-arm MAB stops before the truncation time in each hypothesis, the stopping criterion is met, i.e. the best arm is identified with probability at least $1 - \alpha_j$, so that the power is bound to be close to one whenever $T_j = T(\alpha_j)$.

**mFDR control**   For Fig. 3, we again consider Gaussian draws as in Fig. 2. This time however, for each true null hypothesis we skip the bandit experiment and directly draw $P^j \sim [0,1]$ to compare with the significance levels $\alpha_j$ from our online FDR procedure 2 (see App. B.2 for motivation of this setting). By Theorem 1, mFDR should still be controlled as it only requires the $p$-values to be super-uniform. In Fig. 3(a) we plot the instantaneous false discovery proportion $\text{FDP}(J) = \frac{\sum_{j \in \mathcal{H}_0 J} R_j}{\sum_{j=1}^{T} R_j}$ over the hypothesis index for different runs with the same settings. Apart from initial fluctuations due to the relatively small denominator, observe how the guarantee for the $\text{FDR}(J) = \mathbb{E}\,\text{FDP}(J)$ with the red line showing its empirical value, transfers to the control of each individual run (blue lines).

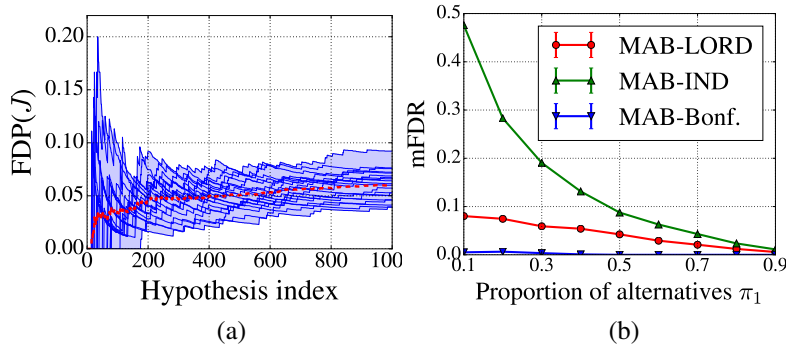

(a)                                                    (b)

Figure 3: (a) Single runs of MAB-LORD (blue) and their average (red) with uniformly drawn $p$-values for null hypotheses and Gaussian draws as in Figure 2. (b) mFDR over different proportions of non-nulls $\pi_1$, with same settings, averaged over 80 runs.

In Figure 3 (b), we compare the mFDR of MAB-FDR against MAB-IND and a Bonferroni type correction. The latter uses a simple union bound and chooses $\alpha_j = \frac{6\alpha}{\pi^2 j^2}$ such that $\sum_{j=1}^{\infty} \alpha_j \leq \alpha$ and thus trivially allows for any time FWER, implying FDR control. As expected, Bonferroni is too conservative and barely makes any rejections whereas the naive MAB-IND approach does not control FDR. LORD avoids both extremes and controls FDR while having reasonable power.

# 5   Discussion

The recent focus in popular media about the lack of reproducibility of scientific results erodes the public's confidence in published scientific research. To maintain credibility of claimed discoveries, simply decreasing the statistical significance levels $\alpha$ of each individual experimental work (e.g., reject at level 0.001 rather than 0.05) would drastically hurt power. A common approach is instead to control the ratio of false discoveries to claimed discoveries at some desired value over many sequential experiments, requiring the statistical significances $\alpha_j$ to change from experiment to experiment. Unlike earlier works on online FDR control, our framework synchronously interacts with adaptive sampling methods like MABs to make the overall sampling procedure per experiment much more efficient than uniform sampling. To the best of our knowledge, it is the first work that successfully combines the benefits of adaptive sampling and FDR control. It is worthwhile to note that any improvement, theoretical or practical, to either online FDR algorithms or best-arm identification in MAB, immediately results in a corresponding improvement for our MAB-FDR framework.

More general notions of FDR with corresponding online procedures have recently been developed by Ramdas et al [14]. In particular, they incorporate the notion of memory and a priori importance of each hypothesis. This could prove to be a valuable extension for our setting, especially in cases when only the percentage of wrong rejections in the recent past matters. It would be useful to establish FDR control for these generalized notions of FDR as well.

**Acknowledgements**

This work was partially supported by Office of Naval Research MURI grant DOD-002888, Air Force Office of Scientific Research Grant AFOSR-FA9550-14-1-001, and National Science Foundation Grants CIF-31712-23800 and DMS-1309356.

## Footnotes

[1]A valid $P^j$ must be stochastically dominated by a uniform distribution on $[0, 1]$, which we henceforth refer to as *super-uniformly distributed*.

[2]Here we have ignored some doubly-logarithmic factors.

[3]The code for reproducing all experiments and plots in this paper is publicly available at https://github.com/fanny-yang/MABFDR

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
