[Supplementary Material]

# A Notation and Figures

| Notation | Terminology and explanation |
|---|---|
| MAB | (pure exploration for best-arm identification in) multi-armed bandits |
| FDR($J$) | the expected ratio of # false discoveries to # discoveries up to experiment $J$ |
| mFDR($J$) | the ratio of expected # false discoveries to expected # discoveries |
| $\alpha$ | target for FDR or mFDR control after any number of experiments |
| BDR($J$) | the best arm discovery rate (generalization of test power) |
| $\epsilon$BDR($J$) | the $\epsilon$-best arm discovery rate (softer metric than BDR) |
| LCB, UCB | the lower and upper confidence bounds used in the best-arm algorithms |
| $j \in \mathbb{N}$ | experiment counter (number of MAB instances) |
| $T_j \in \mathbb{N}$ | stopping time for the $j$-th experiment |
| $P_t^j, P_t \in [0,1]$ | always valid $p$-value after time $t$ (in experiment $j$, explicit or implicit) |
| $P^j$ | always valid $p$-value for experiment $j$ at its stopping time $T_j$ |
| $\alpha_j \in [0,1]$ | threshold set by the online FDR algorithm for $P^j$, using $\{p_i\}_{i=1}^{j-1}$ |
| $T(\alpha_j) \in \mathbb{N}$ | stopping time for the $j$-th experiment, when experiment uses $\alpha_j$ |
| $0$ | the control or default arm |
| $\{1, \ldots, K\}$ | $K = K(j)$ alternatives or treatment arms (experiment $j$ implicit) |
| $i \in \{0, \ldots, K\}$ | $K+1$ options or "all arms" |
| $i_\star, i_b$ | the best of all arms, and the arm returned by MAB |
| $\mu_i, \mu_*$ | the mean of the $i$-th arm, and the mean of the best arm |
| $t, n_i(t) \in \mathbb{N}$ | total number of pulls, number of times arm $i$ is pulled up to time $t$ |

Table 1: Common notation used throughout the paper.

## A.1 Figure: Illustration for the modified LUCB algorithm

In this section we provide intuition how the approximation factor $\epsilon$ affects the commonly known LUCB algorithm.

Figure 4: (a) The means of arms $\{1,2,3\}$ are within $\epsilon$ of the best arm, but only arms $\{1,2\}$ are at least $\epsilon$ better than the control arm 0. Thus, returning any of arms $\{3,4,5\}$ would result in a false discovery when $\epsilon > 0$. (b) An example of the stopping condition being critically met and returning a non-control arm $h_t$. While $\text{LCB}_{h_t} > \text{UCB}_{\ell_t} - \epsilon$ is satisfied with some slack, $\text{LCB}_{h_t} > \text{UCB}_0 + \epsilon$ is just barely satisfied.

# B Experiment details

In this section we provide further detail about the setup and interpretation of our experimental results. We also include plots for artificial Bernoulli draws and the experiments based on the real-world New Yorker Cartoon Caption Contest.

## B.1 Bernoulli and Gaussian draws on artificial data

For the Gaussian draws, we set $\mu_{i_\star} = 8$. The gap to the second best arm is $\Delta = 3$ so that all means $\mu_{i \neq i_\star}$ are drawn uniformly between $Unif \sim [0, 5]$. The number of hypotheses is fixed to be 500. For Bernoulli draws we choose the maximum mean to be $\mu_{i_\star} = 0.4$, $\Delta = 0.3$ so that all means $\mu_{i \neq i_\star}$ are drawn uniformly between $Unif \sim [0, 0.1]$. The number of hypotheses is fixed at 50. We display the empirical average over 100 runs where each run uses the same hypothesis sequence (indicating which hypotheses are true and false) and sequence of means $\mu_i$ for each hypothesis. The only randomness we average over comes from the random Gaussian/Bernoulli draws which cause different rejections $R_j$ and $\alpha_j$, so that the randomness in each draw propagates through the online FDR procedure. The results can be seen in Figures 2 and 5.

Figure 5: (a) Power over truncation time $T_S$ (per hypothesis) for 50 arms and (b) Sample complexity over number of arms for truncation time $T_S = 5000$ for Bernoulli draws with fixed $\mu_{i_\star} = 0.7$, $\Delta = 0.3$ over 50 hypotheses with 20 non-nulls, averaged over 100 runs.

Note that the behavior for both Gaussian and Bernoullis are comparable, which is not surprising due to the choice of the subGaussian LIL bound. However one may notice that the choice of the gap of $\Delta = 3$ vs. $\Delta = 0.3$ drastically increases sample complexity so that the phase transition for power is shifted to very large $T_S$.

### B.1.1 Application to New Yorker captions

In the simulations with real data we consider the crowd-sourced data collected for the *New Yorker Magazine's* Cartoon Caption contest: for a fixed cartoon, captions are shown to individuals online one at a time and they are asked to rate them as 'unfunny', 'somewhat funny', or 'funny'. We considered 30 contests[4] where for each contest, we computed the fraction of times each caption was rated as either 'somewhat funny' or 'funny'. We treat each caption as an arm, but because each caption was only shown a finite number of times in the dataset, we simulate draws from a Bernoulli distribution with the observed empirical mean computed from the dataset. When considering subsets of the arms in any given experiment, we always use the captions with the highest empirical means (i.e. if $n = 10$ then we use the 10 captions that had the highest empirical means in that contest).

Although MAB-FDR still outperforms AB-FDR by a large margin, the plots in Figures 6 also show how the power and sample complexity notably differ from our toy simulation, where we seem to have chosen a rather benign distribution of means - in this setting, the gap $\Delta$ is much lower, often around $\sim 0.01$.

Figure 6: (a) BDR over number of arms, i.e. truncation time per hypothesis for 10 arms and (b) Sample complexity over number of arms for truncation time $T_S = 130000$ for Bernoulli draws, 30 hypotheses with 12 non-nulls and averaged over 100 runs.

## B.2 mFDR and FDR control

In this section we use simulations to demonstrate the second part of our meta algorithm which deals with the control of the false discovery rate or its modified version. Since bandit algorithms have a very high best-arm discovery guarantee which in practice even exceeds its theoretical guarantee of at least $1 - \alpha_j$, mFDR and FDR plots on MAB-FDR directly do not lead to very insightful plots - namely the constant 0 line. However, we can demonstrate that even under adversarial conditions, i.e. when the $P$-value under the null is much less concentrated around one than obtained via the best arm bandit algorithm, mFDR or the false discovery proportion (FDP) in each run are still controlled *at any time* $t$ as Theorem 1 guarantees. Albeit not exactly reflecting mFDR control in the case of MAB-FDR but in fact in an even harder setting, results from these experiments can be regarded as valuable on their own - it emphasizes the fact that Theorem 1 guarantees mFDR control independent of the adaptive sampling algorithm and specific choice of $p$-value as long as it is always valid.

## C  Proof of Proposition 1

For any fixed $\gamma \in (0, 1)$, we have the equivalence
$$\widehat{\mu}_{i,n_i(t)} - \varphi_{n_i(t)}(\tfrac{\gamma}{2K}) > \widehat{\mu}_{0,n_0(t)} + \varphi_{n_0(t)}(\tfrac{\gamma}{2}) + \epsilon \quad \Longleftrightarrow \quad p_{i,t} \leq \gamma.$$
If $\max_{i=1,\dots,K} \mu_i \leq \mu_0 + \epsilon$, then we have

$$\mathbb{P}\left( \bigcup_{i=1}^{K} \bigcup_{t=1}^{\infty} \left\{ \widehat{\mu}_{i,n_i(t)} - \varphi_{n_i(t)}(\tfrac{\gamma}{2K}) > \widehat{\mu}_{0,n_0(t)} + \varphi_{n_0(t)}(\tfrac{\gamma}{2}) + \epsilon \right\} \right)$$

$$= 1 - \mathbb{P}\left( \bigcap_{i=1}^{K} \bigcap_{t=1}^{\infty} \left\{ \widehat{\mu}_{i,n_i(t)} - \varphi_{n_i(t)}(\tfrac{\gamma}{2K}) \leq \widehat{\mu}_{0,n_0(t)} + \varphi_{n_0(t)}(\tfrac{\gamma}{2}) + \epsilon \right\} \right)$$

$$\leq 1 - \mathbb{P}\left( \bigcap_{t=1}^{\infty} \left\{ \mu_0 \leq \widehat{\mu}_{0,t} + \varphi_t(\tfrac{\gamma}{2}) \right\} \cap \bigcap_{i=1}^{K} \bigcap_{t=1}^{\infty} \left\{ \widehat{\mu}_{i,n_i(t)} - \varphi_{n_i(t)}(\tfrac{\gamma}{2K}) \leq \mu_i \right\} \right)$$

$$\leq \mathbb{P}\left( \bigcup_{t=1}^{\infty} \left\{ \mu_0 > \widehat{\mu}_{0,t} + \varphi_t(\tfrac{\gamma}{2}) \right\} \right) + \sum_{i=1}^{K} \mathbb{P}\left( \bigcup_{t=1}^{\infty} \left\{ \widehat{\mu}_{i,n_i(t)} - \varphi_{n_i(t)}(\tfrac{\gamma}{2K}) > \mu_i \right\} \right)$$

$$\leq \tfrac{\gamma}{2} + K \tfrac{\gamma}{2K}$$

$$= \gamma$$

by equation (4). Thus, we have $\mathbb{P}\left( \bigcup_{i=1}^{K} \bigcup_{t=1}^{\infty} \left\{ p_{i,t} \leq \gamma \right\} \right) \leq \gamma$, which completes the proof.

# D   Proof of Proposition 2

We first prove that the algorithm 1 terminates in finite time before moving on to prove the sample complexity statements for the two claims. It suffices to argue for $\delta/2 \leq 0.1$ and we discuss the other case at the end.

## D.1   Proof of termination in finite time

First we prove by contradiction that the algorithm terminates in finite time with probability one for the case $\mu_0 \geq \max_{i=1}^{K} \mu_i - \epsilon$.

Assuming that there exist runs for which the algorithm does not terminate, the set of arms defined by

$$S := \{i : \text{LCB}_0(t) \leq \text{UCB}_i(t) - \epsilon \text{ infinitely often (i.o.)}\}$$

is necessarily non-empty for these runs. We now show that this assumption yields a contradiction so that

$$\mathbb{P}(\text{Algorithm does not terminate}) \leq \mathbb{P}(\text{LCB}_0(t) \leq \max_{i=1,\ldots,K} \text{UCB}_i(t) - \epsilon \text{ i.o.}) = 0 \qquad (11)$$

First take note that by definition of the algorithm, if an arm $i$ is drawn infinitely often (i.o.), then so is the control arm 0 and we have $\text{LCB}_0(t) \to \mu_0$ as well as $\text{UCB}_i(t) \to \mu_i$ as $t \to \infty$. This follows by the law of large numbers combined with the fact that $\varphi_{n_i(t)}, \varphi_{n_0(t)} \to 0$ as $t \to \infty$, since $\varphi_n \to 0$ as $n \to \infty$. Since for the null hypothesis we have $\mu_0 > \mu_i - \epsilon$, it follows that $\text{LCB}_0(t) > \text{UCB}_i(t) - \epsilon$ for all $t \geq t'$ for some $t'$.

This argument implies that all arms $i \in S$ can only be drawn a finite number of times, i.e. $n_i(t) < \infty$ for all $i \in S$. However, the fact that they are not drawn i.o. implies that $h_t \neq i$ and $\ell_t \neq i$ i.o. for all $i \in S$, so that there exists $i' \notin S$ such that $\max_{i \in S} \text{UCB}_i(t) \leq \text{UCB}_{i'}(t)$ i.o. By definition of $S$ we then obtain

$$\text{LCB}_0(t) \leq \text{UCB}_{i'}(t) - \epsilon \text{ i.o.} \qquad (12)$$

However, since $i' \notin S$, inequality (12) cannot hold and equation (11) is proved.

A nearly identical argument to the above shows that the stopping condition is met in finite time.

## D.2   Proof for sample complexity

Define $i_\star = \arg\max_{i=0,1,\ldots,K} \mu_i$ (breaking ties arbitrarily) and $n_i(t)$ to be the number of times sample $i$ was drawn until time $t$. For any $i \in \{0, 1, \ldots, K\}$ and $\eta \in \mathbb{R}$ we define the following key quantity

$$\tau_i(\eta, \xi) := \min\{n \in \mathbb{N} : 2\varphi_n(\tfrac{\delta}{2K}) < \max\{|\eta - \mu_i|, \xi\}\} \qquad (13)$$
$$\lesssim \min\left\{(\eta - \mu_i)^{-2} \log(K \log((\eta - \mu_i)^{-2})/\delta), \xi^{-2} \log(K \log(\xi^{-2})/\delta)\right\}$$

where we set $\tau_i(\mu_i, 0) = \infty$, but this case will not arise because whenever $\eta = \mu_i$ for some $i$, we will necessarily have $\xi > 0$.

Let us define the events

$$\mathcal{E}_i = \bigcap_{n=1}^{\infty} \{|\widehat{\mu}_{i,n} - \mu_i| \leq \varphi_n(\tfrac{\delta}{2K})\}.$$

By a union bound and the LIL bound in (4), we have for $\delta/2K < 0.1$ that $\mathbb{P}\left(\bigcup_{i=0}^{K} \mathcal{E}_i^c\right) \leq \frac{K+1}{2K}\delta \leq \delta$ for $K \geq 2$. For $\frac{\delta}{2K} > 0.1$, note that for all $\delta' < \delta$ we have $\varphi_n(\delta') \leq \varphi_n(\delta)$ so that

$$\mathbb{P}(\mathcal{E}_i^c) = \mathbb{P}(\varphi_n(\tfrac{\delta}{2K}) < \widehat{\mu}_{i,n} - \mu_i)$$
$$\leq \mathbb{P}(\varphi_n(0.1) < \widehat{\mu}_{i,n} - \mu_i) \leq \tfrac{\delta}{2K} \qquad \forall i = 1, \ldots, K$$

Throughout the rest of the proof we assume the events $\mathcal{E}_i$ hold.

The following simple lemma regarding the key quantity $\tau_i$ will be used throughout the proof.

**Lemma 1.** *Fix $i \in \{0, 1, \ldots, K\}$ and $\eta > 0$. For any $t \in \mathbb{N}$, whenever $n_i(t) \geq \tau_i(\eta, \xi)$ we have that under the event $\bigcap_{i=0,\ldots,K} \mathcal{E}_i$, we have*

$$UCB_i(t) \leq \max\{\eta, \mu_i + \xi\} \text{ if } \eta \geq \mu_i$$
$$LCB_i(t) \geq \min\{\eta, \mu_i - \xi\} \text{ if } \eta \leq \mu_i$$

*Proof.* Assume $n_i(t) \geq \tau_i(\eta, \xi)$. If $\eta \geq \mu_i$ we have by definition of $\mathcal{E}_i$ that

$$\text{UCB}_i(t) = \widehat{\mu}_{i,n_i(t)} + \varphi_{n_i(t)}(\tfrac{\delta}{2}) \leq \mu_i + 2\varphi_{n_i(t)}(\tfrac{\delta}{2K}) < \mu_i + \max\{\eta - \mu_i, \xi\}$$

and if $\eta \leq \mu_i$

$$\text{LCB}_i(t) = \widehat{\mu}_{i,n_i(t)} - \varphi_{n_i(t)}(\tfrac{\delta}{2K}) \geq \mu_i - 2\varphi_{n_i(t)}(\tfrac{\delta}{2K}) > \mu_i - \max\{\mu_i - \eta, \xi\} = \mu_i + \min\{\eta - \mu_i, -\xi\}$$

$\square$

### D.2.1 Proof of Proposition 2 (a) $\mu_0 > \max\limits_{i=1,\ldots,K} \mu_i - \epsilon$

At each time $t$ which does not satisfy the stopping condition, arm 0 and $\arg\max_{i=1,\ldots,K} \text{UCB}_i(t)$ are pulled. Note that by Lemma 1

$$\{n_0(t) \geq \tau_0(\tfrac{\mu_0 + (\max_{i=1,\ldots,K} \mu_i - \epsilon)}{2}, 0)\} \implies \text{LCB}_0(t) \geq \min\{\tfrac{\mu_0 + (\max_{i=1,\ldots,K} \mu_i - \epsilon)}{2}, \mu_0\} \geq \tfrac{\mu_0 + (\max_{i=1,\ldots,K} \mu_i - \epsilon)}{2} \tag{14}$$

so that $t > n_0(t)$ makes sure that there were enough draws for the particular arm 0 (since it's drawn every time). For $i \neq 0$ we have

$$\{n_i(t) \geq \tau_i(\tfrac{(\mu_0 + \epsilon) + \max_{i=1,\ldots,K} \mu_i}{2}, 0)\} \implies \text{UCB}_i(t) \leq \max\{\tfrac{(\mu_0 + \epsilon) + \max_{i=1,\ldots,K} \mu_i}{2}, \mu_i\} \leq \tfrac{(\mu_0 + \epsilon) + \max_{i=1,\ldots,K} \mu_i}{2}. \tag{15}$$

which makes $t > \sum_{i=0}^{K} n_i(t)$ a necessary condition.

Reversely whenever $t > \sum_{i=0}^{K} n_i(t)$, for all arms $i \neq 0$ we have $\text{UCB}_i(t) \leq \tfrac{(\mu_0 + \epsilon) + \max_{i=1,\ldots,K} \mu_i}{2}$. In essence, once arm $i$ has sampled $n_i(t)$ times, because of (15), it will not be sampled again - either, because all of the other $UCB_i(t)$ satisfy the same upper bound, the algorithm will have stopped, or, if for some $i$ we have $UCB_i(t) > \tfrac{(\mu_0 + \epsilon) + \max_{i=1,\ldots,K} \mu_i}{2}$ that will be the arm that is drawn. Thus,

$$\{t \geq B_1(\mu, \delta) := \tau_0(\tfrac{\mu_0 + (\max\{\max_{i=1,\ldots,K} \mu_i - \epsilon\})}{2}, 0) + \sum_{i=1}^{K} \tau_i(\tfrac{(\mu_0 + \epsilon) + \max_{i=1,\ldots,K} \mu_i}{2}, 0)\}$$

$$\implies \{\text{LCB}_0(t) - \text{UCB}_i(t) \geq -\epsilon \quad \forall i \neq 0\},$$

i.e., the stopping condition is met, where the first term accounts for satisfying (14), the second term accounts for satisfying (15) for all $i \neq 0$, and the third term accounts for satisfying Equation (16). Denoting $T(\delta)$ as the stopping time of the algorithm, this implies that with probability at least $1 - \delta$, we have $T(\delta) \leq B_1(\mu, \delta)$ and arm 0 is returned.

Let us now simplify the expression to make it more accessible to the reader and arrive at the theorem statement. Defining $\max\{|\eta - \mu_i|, \xi\}$ as the *effective gap* in the definition of $\tau_i(\eta, \xi)$ in Equation 13, it is straightforward to verify that the effective gap associated with arm 0 is equal to

$$\alpha_0 \gtrsim \min\left\{(\mu_0 + \epsilon) - \max_{j=1,\ldots,K} \mu_j\right\}.$$

And the effective gap for any other arm $i$ is equal to

$$\alpha_i \gtrsim \min\left\{(\mu_0 + \epsilon) - \mu_i\right\}.$$

Using these quantities, we can see that the upper bound $B_1(\mu, \delta)$ scales like $\sum_{i=0}^{K} \alpha_i^{-2} \log(K \log(\alpha_i^{-2})/\delta)$.

**D.2.2 Proof of Proposition 2 (b)** $\max_{i=1,\ldots,K} \mu_i = \mu_{i_\star} > \mu_0 + \epsilon$

At each time $t$ which does not satisfy the stopping condition, arm 0 is pulled. Note again that by Lemma 1

$$\{n_0(t) \geq \tau_0(\tfrac{(\mu_{i_\star}-\epsilon)+\mu_0}{2}, 0)\} \implies \text{UCB}_0(t) \leq \max\{\tfrac{(\mu_{i_\star}-\epsilon)+\mu_0}{2}, \mu_0\} \leq \frac{(\mu_{i_\star}-\epsilon)+\mu_0}{2}.$$

The following claim is key to proving this case (where $u \in (0,1)$ be an absolute constant to be defined later).

**Claim 1.** *Under the event $\bigcap_{i=0,\ldots,K} \mathcal{E}_i$, for any $u \leq \tfrac{2}{7}$ and $\bar\mu \in [\max_{j \neq i_\star} \mu_j, \mu_{i_\star}]$, we have*

$$|\{s \geq 2\sum_{i=0}^K \tau_i(\bar\mu, u\epsilon) : LCB_{h_s}(s) \leq \mu_{i_\star} - \tfrac{5}{2}u\epsilon \text{ or } UCB_{\ell_s}(s) \geq \mu_{i_\star} + u\epsilon\}| < \sum_{i=0}^K \tau_i(\bar\mu, u\epsilon) \quad (16)$$

The proof of this claim can be found in Appendix G. Note that for all $s$ we have that

$$\text{LCB}_{h_s}(s) \geq \mu_{i_\star} - \tfrac{5}{2}u\epsilon \text{ and } \text{UCB}_{\ell_s}(s) \leq \mu_{i_\star} + u\epsilon \implies \text{LCB}_{h_s}(s) \geq \text{UCB}_{\ell_s}(s) - \epsilon.$$

Intuitively the inequality (16) thus limits the number of times that for $t \geq 2\sum_{i=0}^K \tau_i(\bar\mu, u\epsilon)$, the criterion $\text{LCB}_{h_s}(s) \geq \text{UCB}_{\ell_s}(s) - \epsilon$ is not fulfilled. We refer to the times when it is fulfilled, as "good" times.

Applying Claim 1 with $\bar\mu = \max_{j \neq i_\star} \frac{\mu_{i_\star}+\mu_j}{2}$ and $u = \frac{\mu_{i_\star}-(\mu_0+\epsilon)}{5\epsilon}$ we then observe that on the "good" times, we have

$$\text{LCB}_{h_t} \geq \mu_{i_\star} - \tfrac{5}{2}u\epsilon = \frac{\mu_{i_\star}+(\mu_0+\epsilon)}{2} = \frac{(\mu_{i_\star}-\epsilon)+\mu_0}{2} + \epsilon,$$

so that we directly obtain that with probability at least $1 - \delta$,

$$T(\delta) \leq B_2(\mu, \delta) := \tau_0(\tfrac{(\mu_{i_\star}-\epsilon)+\mu_0}{2}, 0) + 3\sum_{i=0}^K \tau_i(\max_{j \neq i_\star} \tfrac{\mu_{i_\star}+\mu_j}{2}, \min\{\tfrac{2}{7}\epsilon, \tfrac{\mu_{i_\star}-(\mu_0+\epsilon)}{5}\}).$$

Let us now simplify the expression. It is straightforward to verify that the effective gap associated with arm 0 is equal to

$$\alpha_0 \gtrsim \min\left\{ \frac{(\max_{i=1,\ldots,K} \mu_i - \epsilon) - \mu_0}{2}, \max\left\{ \max_{j \neq i_\star} \tfrac{\mu_{i_\star}+\mu_j}{2} - \mu_0, \tfrac{2}{7}\epsilon \right\} \right\}$$

$$\gtrsim \min\left\{ \max_{j=1,\ldots,K} \mu_j - (\mu_0+\epsilon), \max\{\Delta_0, \epsilon\} \right\}$$

and the effective gap for any other arm $i$ is equal to

$$\alpha_i \gtrsim \max\left\{ |\max_{j \neq i_\star} \tfrac{\mu_{i_\star}+\mu_j}{2} - \mu_i|, \min\{\tfrac{2}{7}\epsilon, \frac{(\max_{i=1,\ldots,K} \mu_i - \epsilon) - \mu_0}{5}\} \right\}$$

$$\gtrsim \max\left\{ \Delta_i, \min\left\{ \max_{j=1,\ldots,K} \mu_j - (\mu_0+\epsilon), \epsilon \right\} \right\}$$

where we recall that $\Delta_i = \mu_{i_\star} - \mu_i$ if $i \neq i_\star$, and $\Delta_{i_\star} = \mu_{i_\star} - \max_{j \neq i_\star} \mu_j$ otherwise. Using these quantities, the upper bound $B_2(\mu, \delta)$ on the stopping time $T(\delta)$ scales like $\sum_{i=0}^K \alpha_i^{-2} \log(K \log(\alpha_i^{-2})/\delta)$. This concludes the proof of the proposition.

# E  Proof of Theorem 1

We now turn to the proof of Theorem 1, splitting our argument into parts (a) and (b), respectively.

### E.1 Proof of part (a)

In order for generalized alpha-investing procedures such as LORD to successfully control the mFDR, it is sufficient that $p$-values under the null be *conditionally super-uniform*, meaning that for all $j \in \mathcal{H}_0$, we have

$$\mathbb{P}_0(P^j \leq \alpha_j | \mathcal{F}^{j-1}) \leq \alpha_j(R_1, \ldots, R_{j-1}) \tag{17}$$

where $\mathcal{F}^{j-1}$ is the $\sigma$-field induced by $R_1, \ldots, R_{j-1}$. Note that as long as condition (17) is satisfied, $T_j$ and thus $P^j$ could potentially depend on $\alpha_j$, i.e. the rejection indicator variables $R_1, \ldots, R_{j-1}$ and potentially $P^1, \ldots, P^{j-1}$. See Aharoni and Rosset [3] for further details.

It thus suffices to show that condition (17) holds for our definition of $p$-value in our framework. We know that by Proposition 1 we have for any random stopping time, thus any fixed truncation time $T_S$, that $\mathbb{P}_0(P_T^j \leq \alpha_j) \leq \alpha_j$. We now show that the same bound also holds for the ($\alpha_j$-dependent) bandit stopping time $T(\alpha_j)$, i.e. that $\mathbb{P}_0(P_{T(\alpha_j)}^j \leq \alpha_j) \leq \alpha_j$.

Under the null hypothesis, the best arm is at most $\epsilon$ better than the control arm, i.e. $\mu_0 > \mu_i - \epsilon$, so that by Proposition 2 we have that with probability $\geq 1 - \alpha_j$, $i_b = 0$, i.e. $\text{LCB}_0(t) > \text{UCB}_i(t) - \epsilon$ for all $i \neq 0$. Hence, $\text{LCB}_i(t) - \text{UCB}_0(t) < \epsilon$, and thus, by the definition of the $p$-values, $P_{i,T(\alpha_j)}^j = 1$ for all $i$ with probability $\geq 1 - \alpha_j$. It finally follows that $\mathbb{P}_0(P_{T(\alpha_j)}^j \leq \alpha_j) \leq \alpha_j$.

Putting things together, under the true null hypothesis (omitting the index $j \in \mathcal{H}_0$ to simplify notation) we directly have that for any $\alpha_j$

$$
\begin{aligned}
\mathbb{P}_0(P_{T_j}^j(\alpha_j) \leq \alpha_j) &= \mathbb{P}_0\big(P_{T(\alpha_j)}^j \leq \alpha_j \big| T(\alpha_j) \leq T_S\big) \mathbb{P}_0(T(\alpha_j) \leq T_S) \\
&\quad + \mathbb{P}_0\big(P_{T_S}^j \leq \alpha_j \big| T(\alpha_j) > T_S\big) \mathbb{P}_0(T(\alpha_j) > T_S) \\
&\leq \alpha_j(\mathbb{P}_0(T(\alpha_j) \leq T_S) + \mathbb{P}_0(T(\alpha_j) > T_S)) = \alpha_j
\end{aligned}
$$

for all fixed $\alpha_j$ even when the stopping time $T(\alpha_j)$ is dependent on $\alpha_j$. This is equivalent to stating that for any sequence $R_1, \ldots, R_{j-1}$ we have

$$
\begin{aligned}
\mathbb{P}_0(P^j \leq \alpha_j(R_1, \ldots, R_{j-1}) | \mathcal{F}^{j-1}) &= \mathbb{P}_0(P_{T(\alpha_j(R_1, \ldots, R_{j-1}))}^j \leq \alpha_j(R_1, \ldots, R_{j-1})) \\
&\leq \alpha_j(R_1, \ldots, R_{j-1})
\end{aligned}
$$

and the proof is complete.

### E.2 Proof of part (b)

It suffices to prove that for a single experiment $j$ and $T_S = \infty$, we have $\mathbb{P}_1(P_{T(\alpha_j)}^j \leq \alpha_j) \geq 1 - \alpha_j$ where $\mathbb{P}_1$ is the distribution of a non-null experiment $j$. First observe that at stopping time $T(\alpha_j)$ of Algorithm 1, either $P_{i,T(\alpha_j)}^j \leq \alpha_j$ or $P_{i,T(\alpha_j)}^j = 1$ for all $i$. The former event happens whenever the algorithm exits with $i_b \in \mathcal{S}^\star$, i.e. when $\text{LCB}_{i_b}(t) \geq \text{UCB}_{\ell_t}(t) - \epsilon$ holds. Then, by definition of the $p$-value in equation (6) and $\ell_t$ we must have that $P_{i_b,T(\alpha_j)}^j \leq \alpha_j$. As a consequence, by Proposition 2, we have

$$
\begin{aligned}
\mathbb{P}_1(P_{T(\alpha_j)}^j \leq \alpha_j) &\geq \mathbb{P}(P_{T(\alpha_j)}^j \leq \alpha_j) \\
&\geq \mathbb{P}_1(\text{Algorithm 1 exits with } i_b \in \mathcal{S}^\star) \\
&\geq 1 - \alpha_j.
\end{aligned}
$$

## F  Notes on FDR control

We can prove FDR control for our framework using the specific online FDR procedure called LORD '15 introduced in [13]. When used in Procedure 2, the only adjustment that needs to be made is to reset $W(j+1)$ to $\alpha$ in step 2 after every rejection, yielding $\alpha_j = \alpha \gamma_{j-\tau_j}$ for any sequence $\{\gamma_j\}_{j=1}^\infty$ such that $\sum_{j=1}^\infty \gamma_j = 1$. We call the adjusted procedure MAB-LORD' for short.

**Theorem 2** (Online FDR control for MAB-LORD). *(a) MAB-LORD' achieves mFDR and FDR control at a specified level $\alpha$ for stopping times $T_j = \min\{T(\alpha_j), M\}$.*

*(b) Furthermore, if we set $T_S = \infty$, MAB-LORD' satisfies*

$$\epsilon BDR(J) \geq \frac{(1-\alpha)}{|\mathcal{H}_1(J)|}. \tag{18}$$

Note that LORD as in [13] is less powerful than in [4] since the values $\alpha_j$ in the former can be much smaller than those in [4], which could in fact exceed the level $\alpha$. Therefore, for FDR control we currently do have to sacrifice some power.

*Proof.* We leverage the proposition that can be obtained from a slightly more careful analysis of the procedure than in [13].

**Proposition 3.** *If $\mathbb{P}_0(P^j \leq \alpha_j \mid \tau_j) \leq \alpha_j$, i.e. the distribution of the $p-$values under the null are superuniform conditioned on the last rejection, using the online LORD'15 procedure controls the FDR at each $t$.*

Note that this proposition allows online FDR control for any, possibly dependent, $p$-values which are conditionally superuniform. This condition is not equivalent to (17) in general, it is in fact less restrictive since the probability is conditioned only on a function $\tilde{\tau}_j = \max\{k \leq j : R_k = 1\}$ of all past rejections. Formally, the sigma algebra induced by $\tau_{j-1}$ is contained in $\mathcal{F}^{j-1}$ and hence $\mathbb{P}_0(P^j \leq \alpha_j \mid \tau_{j-1}) \leq \mathbb{P}_0(P^j \leq \alpha_j \mid R_1, \ldots, R_j)$ by the tower property. Finally, utilizing the fact that our $p$-values are conditionally super-uniform as proven in Section E.1, i.e. inequality (17) holds, the condition for Proposition 3 is fulfilled and the proof is complete. □

## F.1 Proof of Proposition 3

Let $\tilde{\tau}_i$ denote the time of the $i$-th rejection with $\tilde{\tau}_0 = 0$ (note that this is different from $\tau_j$). and define $k(t) = \sum_{j=1}^{t} R_j$. Let $H_j$ be the $j-$th hypothesis that was rejected. We adjust an argument from [13].

First observe that $\{k(t) = \ell\} = \{\tilde{\tau}_\ell \leq t, \tilde{\tau}_{\ell+1} > t\}$ and $FDP(t) = FDP(\tilde{\tau}_{k(t)})$ so that

$$\mathbb{E}FDP(t) = \mathbb{E}FDP(\tau_{k(t)}) = \sum_{\ell=1}^{t} \mathbb{E}\Big[\frac{\sum_{j\in\mathcal{H}_0} R_j}{\ell} \mid k(t) = \ell\Big] P(k(t) = \ell)$$

$$= \sum_{\ell=1}^{t} P(k(t) = \ell) \sum_{i=1}^{\ell} \mathbb{E}\Big[\frac{\mathbb{1}_{H_i \in \mathcal{H}_0}}{\ell} \mid k(t) = \ell\Big]$$

$$= \sum_{\ell=1}^{t} P(k(t) = \ell) \sum_{i=1}^{\ell} \mathbb{E}\Big[\mathbb{E}\Big(\frac{\sum_{j=\tilde{\tau}_{i-1}+1}^{\tilde{\tau}_i} R_j \mathbb{1}_{j\in\mathcal{H}_0}}{\ell} \mid \tilde{\tau}_0, \ldots, \tilde{\tau}_{i-1}\Big) \mid \tilde{\tau}_\ell \leq t, \tilde{\tau}_{\ell+1} > t\Big]$$

Since for the LORD '15 procedure, we have $\alpha_t = \gamma_{t-\tau_t}$, and thus for all positive integers $i$, the random variables $R_j$ with $j \geq \tilde{\tau}_{i-1}$ are conditionally independent of $\tilde{\tau}_0, \ldots, \tilde{\tau}_{i-2}$ given $\tilde{\tau}_{i-1}$. Additionally noting that $\tilde{\tau}_{i-1} = \tau_j$ for all $j \geq \tilde{\tau}_{i-1}$ by definition of $\tilde{\tau}$ and $\tau$, using $\mathbb{E}_0(\mathbb{1}_{p_j \leq \alpha_j} \mid \tau_j) \leq \alpha_j$ we obtain

$$\mathbb{E}\Big(\frac{\sum_{j\in(\tilde{\tau}_{i-1}, \tilde{\tau}_i]\cap j\in\mathcal{H}_0} R_j}{\ell} \mid \tilde{\tau}_0, \ldots, \tilde{\tau}_{i-1}\Big) = \mathbb{E}\Big(\frac{\sum_{j=\tilde{\tau}_{i-1}+1}^{\tilde{\tau}_i} R_j \mathbb{1}_{j\in\mathcal{H}_0}}{\ell} \mid \tilde{\tau}_{i-1}\Big)$$

$$\leq \frac{\sum_{j=\tau_{i-1}+1}^{\tau_i} \mathbb{1}_{j\in H_0}\mathbb{E}[R_j \mid \tau_j]}{\ell}$$

$$\leq \frac{\sum_{j=\tau_{i-1}+1}^{\tau_i} \alpha_j}{\ell} \leq \frac{\alpha}{\ell}.$$

The last inequality follows since between any two rejection times $\tau_k, \tau_{k+1}$, we have

$$\sum_{i=\tau_k}^{\tau_{k+1}} \alpha_i \leq \alpha \sum_{i=1}^{\infty} \gamma_i \leq \alpha.$$

Since $\sum_{\ell=1}^{t} P(k(t) = \ell) = 1$ it follows that FDR control is obtained.

# G   Proof of Claim 1

Let $\bar{\mu} \in [\max_{j \neq i_\star} \mu_j, \mu_{i_\star}]$ and $\tau_i := \tau_i(\bar{\mu}, u\epsilon)$. The following result is a a key ingredient for the proof of the claim.

**Proposition 4.** *For any time $t$ and $u \leq 1/2$,*

$$\left\{ |\{s \leq t : h_s = i_\star\}| \geq \sum_{i=0}^{K} \tau_i \right\}$$
$$\implies \{UCB_{\ell_t}(t) \leq \bar{\mu} + u\epsilon\} \cap \{LCB_{h_t}(t) \geq \bar{\mu} - u\epsilon\}$$
$$\implies \{LCB_{h_t}(t) - UCB_{\ell_t}(t) \geq -\epsilon\}.$$

*Proof.* If $h_s = i_\star$ then *some* $i \neq i_\star$ is assigned to $\ell_s$ and $\text{UCB}_i(s) \leq \max\{\bar{\mu}, \mu_i + u\epsilon\} \leq \bar{\mu} + u\epsilon$ whenever $n_i(s) \geq \tau_i(\bar{\mu}, u\epsilon)$. Because $\ell_s$ is the highest upper confidence bound, the sum over all $\tau_i$ represents exhausting all arms (i.e., pigeonhole principle). An analogous result holds for $\text{LCB}_{i_\star}(t)$.   $\square$

A direct consequence of Proposition 4 is that even though we don't know which arm will be assigned to $h_t$ at any given time $t$, we do know that if $h_t = i_\star$ for a sufficient number of times, namely $\sum_{i=0}^{K} \tau_i$ times, the termination criteria will be met. Thus, assume $h_t \neq i_\star$ and note that

$$\{h_t = i, \mu_i < \mu_{i_\star} - \tfrac{5}{2}u\epsilon, \widehat{\mu}_{i,n_i(t)} \geq \min\{\bar{\mu}, \mu_{i_\star} - \tfrac{3}{2}u\epsilon\}\}$$
$$\implies \min\{\bar{\mu}, \mu_{i_\star} - \tfrac{3}{2}u\epsilon\} \leq \widehat{\mu}_{i,n_i(t)} \leq \mu_i + \varphi_{n_i(t)}(\tfrac{\delta}{2K})$$
$$\implies \{n_i(t) < \tau_i\}$$

where the last line follows from $\mu_i + \varphi_{n_i(t)}(\tfrac{\delta}{2K}) < \min\{\bar{\mu}, \mu_i + u\epsilon\} \leq \min\{\bar{\mu}, \mu_{i_\star} - \tfrac{3}{2}u\epsilon\}$ whenever $n_i(t) \geq \tau_i$. Furthermore, the following Proposition 5, says for $t \geq 2\sum_{i=0}^{K} \tau_i$ we have that $\widehat{\mu}_{h_t,n_{h_t}(t)} \geq \min\{\bar{\mu}, \mu_{i_\star} - \tfrac{3}{2}u\epsilon\}$.

**Proposition 5.** *For any time $t$,*

$$\{t \geq 2\sum_{i=0}^{K} \tau_i\} \implies \{\widehat{\mu}_{h_t,n_{h_t}(t)} \geq \min\{\bar{\mu}, \mu_{i_\star} - \tfrac{3}{2}u\epsilon\}\}.$$

The proof of the proposition can be found in Section G.1.

Combining this fact with the display immediately above and the observation that some $i = h_t$, we have that $|\{s \geq 2\sum_{i=0}^{K} \tau_i : \mu_{i_\star} - \mu_{h_s} \geq \tfrac{5}{2}u\epsilon\}| < \sum_{i=0}^{K} \tau_i$. Now, on one of these times $t$ such that $\{h_t = i, n_i(t) \geq \tau_i, \mu_{i_\star} - \mu_i < \tfrac{5}{2}u\epsilon\}$, we have

$$\text{LCB}_i(t) = \widehat{\mu}_{i,n_i(t)} - \varphi_{n_i(t)}(\tfrac{\delta}{2K}) \geq \mu_i - 2\varphi_{n_i(t)}(\tfrac{\delta}{2K}) \geq \min\{\bar{\mu}, \mu_i - u\epsilon\} \geq \mu_{i_\star} - \tfrac{5}{2}u\epsilon.$$

The above display with the next proposition completes the proof of Equation 16.

**Proposition 6.** *For any time $t$,*

$$\{t \geq \sum_{i=0}^{K} \tau_i\} \implies \{\max_{i=0,1,\dots,K} UCB_i(t) \leq \mu_{i_\star} + u\epsilon\}.$$

*Proof.* Note that

$$\{\text{UCB}_i(t) \geq \mu_{i_\star} + u\epsilon\} \implies \{\mu_{i_\star} + u\epsilon \leq \text{UCB}_i(t) = \widehat{\mu}_{i,n_i(t)} + \varphi_{n_i(t)}(\tfrac{\delta}{2}) \leq \mu_i + 2\varphi_{n_i(t)}(\tfrac{\delta}{2K})\}$$
$$\implies \{n_i(t) < \tau_i\}$$

since $\mu_i + 2\varphi_{n_i(t)}(\tfrac{\delta}{2K}) < \max\{\bar{\mu}, \mu_i + u\epsilon\} \leq \mu_{i_\star} + u\epsilon$ whenever $n_i(t) \geq \tau_i$. Now, because at each time $t$, the arm $\arg\max_{j=0,1,\dots,K} \text{UCB}_j(t)$ is pulled because it is either $h_t$ or $\ell_t$, we conclude that this arm can only be pulled $\tau_i$ times before satisfying $\text{UCB}_i(t) \leq \mu_{i_\star} + u\epsilon$.   $\square$

## G.1 Proof of Proposition 5

The above proposition implies,

$$\{t \geq 2\sum_{i=0}^{K}\tau_i\} \implies \left\{|\{s \leq t : h_s \neq i_\star\}| \geq \sum_{i=0}^{K}\tau_i\right\}.$$

Now consider the event

$$\{h_t \neq i_\star, \ell_t = i\} \implies \mu_{i_\star} \leq \widehat{\mu}_{i_\star,n_{i_\star}(t)} + \varphi_{n_{i_\star}(t)}(\tfrac{\delta}{2}) \leq \widehat{\mu}_{i,n_i(t)} + \varphi_{n_i(t)}(\tfrac{\delta}{2}) \leq \mu_i + 2\varphi_{n_i(t)}(\tfrac{\delta}{2K})$$

$$\implies \{\mu_{i_\star} - \mu_i \leq 2\varphi_{n_i(t)}(\tfrac{\delta}{2K})\}$$

$$\implies \{n_i(t) < \tau_i\} \cup \{n_i(t) \geq \tau_i, \mu_{i_\star} - \mu_i \leq 2\varphi_{n_i(t)}(\tfrac{\delta}{2K})\}$$

$$\implies \{n_i(t) < \tau_i\} \cup \{n_i(t) \geq \tau_i, \mu_{i_\star} - \mu_i \leq \max\{|\bar{\mu} - \mu_i|, u\epsilon\}\}$$

$$\implies \{n_i(t) < \tau_i\} \cup \{n_i(t) \geq \tau_i, \mu_{i_\star} - \mu_i < u\epsilon\} \cup \{n_i(t) \geq \tau_i, i = i_\star\}$$

by the definition of $\tau_i$. Because at each time $s \leq t$ we have that *some* $i = \ell_s$, if $|\{s \leq t : h_s \neq i_\star\}| \geq \sum_{i=0}^{K}\tau_i$, we have that

$$\{t \geq 2\sum_{i=0}^{K}\tau_i\} \implies \{\exists i : n_i(t) \geq \tau_i \text{ and } \mu_{i_\star} - \mu_i < u\epsilon\} \cup \{n_i(t) \geq \tau_i \text{ and } i = i_\star\}.$$

We use the fact that such an $\ell_t = i \neq i_\star$ exists that satisfies $\mu_{i_\star} - \mu_i < u\epsilon$ to say

$$\exists i \neq i_\star : \widehat{\mu}_{i,n_i(t)} \geq \mu_i - \varphi_{n_i(t)}(\tfrac{\delta}{2K}) \geq \mu_i - \max\{\mu_{i_\star} - \mu_i, u\epsilon\}/2 \geq \mu_{i_\star} - \tfrac{3}{2}u\epsilon$$

or $\ell_t = i_\star$ and

$$\widehat{\mu}_{i_\star,n_{i_\star}(t)} \geq \mu_{i_\star} - \varphi_{n_{i_\star}(t)}(\tfrac{\delta}{2K}) \geq \mu_{i_\star} - \max\{\mu_{i_\star} - \bar{\mu}, u\epsilon\}/2 = \min\{\bar{\mu}, \mu_{i_\star} - \tfrac{1}{2}u\epsilon\}.$$

Because $\widehat{\mu}_{h_t,n_{h_t}(t)} \geq \max_{i=0,1,\dots,K}\widehat{\mu}_{i,n_i(t)}$, the proof of the claim is complete.

## Footnotes

[4]Contest numbers 520-551, excluding 525 and 540 as they were not present. Full dataset and its description is available at `https://github.com/nextml/NEXT-data/`.