[Reviews · NeurIPS 2017]

Reviewer 1



The paper looks at continuous improvement using a sequence of A/B tests, and proposes instead to implement adaptive testing such as multi-armed bandit problem while controlling the false discovery rate. This is an important problem discussed in statistical literature, but still unsolved. The approach proposed in this paper seems to apparently solve the issues. This is a very interesting paper, that despite minor concerns listed below, could lead to a potentially new avenue of research. Lines 24-37: There are well-known issues, and it would be desirable to add citations. Although authors clearly focus on CS/ML literature, there is also a relevant body of literature in biometrics, see e.g. survey by Villar, Bowden and Wason (Statistical Science 2015), the references therein and the more recent papers citing this survey. Line 37: "testing multiple literature" -> "multiple testing literature" Line 38-47: A similar concept exists in biometrics, called "platform trials" - please describe how your concept differs Line 112: "and and" -> "and" Line 115: please provide reference for and description of LUCB Line 153: "samplesas" -> "samples as" Line 260: "are ran" -> ? Line 273: It is not clear what "truncation time" is and why it is introduced - it seems to have a huge effect on the results in Figure 2 Line 288-291: While this motivation is interesting, it seems to be mentioned at an inappropriate place in the paper - why not to do it in the introduction, alongside website management and clinical trials?

Reviewer 2



If no proper correction is applied, the repetition of individually well-grounded high-confidence tests is known to lead irremediably to absurd "statistically certified" discoveries. False discovery rate control (FDR) is an alternative of family-wise test corrections like Bonferroni's which are known to be too conservative. This paper propose a general framework for repeated A/B/n tests which integrates several concrete ideas: - introduce a "best-arm identification with control" variant of the (epsilon,delta)-PAC best-arm identification problem that we could also name "better-arm identification"; - replace inefficient static A/B/n testing procedures by adaptive PAC "better-arm" algorithms; - propose a variant of the LUCB algorithm for this purpose; - integrate anytime p-values calculus for continuous FDR control. The proposed "better-arm identification" algorithm is analyzed in section 3.2. The proposed meta-procedures are analyzed and shown to guarantee this FRD control. Some experiments are provided at the end of the paper. At first sight I was a bit surprised by this strongly application-oriented cross-discipline paper, but I really liked the fresh ideas and the concrete perspectives they give for MABs both on the applicative and theoretic grounds. Typo: l37 "testing multiple" -> "multiple testing " l153 "samplesas" -> "samples as"

Reviewer 3



The paper studies sequential hypothesis testing problems, where the decision maker processes hypotheses sequentially (as opposed to processing them jointly). The objective is to control false discovery rate (FDR) or modified FDR (mFDR), and also keep the best arm discovery rate (BDR) high. The proposed algorithm is based on a combination of pure-exploration multi-armed bandit (MAB) methods and results from online FDR control. Using an appropriate pure exploration method ensures that BDR is high, while the online FDR method ensure low FDR. The main contribution of the paper is an appropriate combination of these two components that guarantees simultaneous control of FDR and BDR. Overall, the paper is well-written and the ideas are very clear and easy to follow. The technical contribution is interesting and relevant to NIPS community.